# Dynamic SOFA score assessments to predict outcomes after acute admission of octogenarians to the intensive care unit

Emmanuelle Loyrion[1], Lydiane Agier[2], Thibaut Trouve-Buisson[1], Gaetan Gavazzi[3], Carole Schwebel[4], Jean-Luc Bosson[2], Jean-François Payen[1] *

**1** Department of Anesthesia and Critical Care, CHU Grenoble Alpes, University Grenoble Alpes, Grenoble, France, **2** Department of Public Health, CHU Grenoble Alpes, University Grenoble Alpes, Grenoble, France, **3** Department of Geriatrics, CHU Grenoble Alpes, University Grenoble Alpes, Grenoble, France, **4** Department of Medical Intensive Care, CHU Grenoble Alpes, University Grenoble Alpes, Grenoble, France

* jfpayen@univ-grenoble-alpes.fr

## Abstract

### Background

Identifying which octogenarians could benefit most from continuing critical care is challenging. We aimed to see if responses to therapies using the sequential organ failure assessment (SOFA) score on day 4 after unplanned admission to the intensive care unit (ICU) could be associated with short-term mortality.

### Methods

In this prospective observational cohort study, data from 4 ICUs in a University Hospital included SOFA scores on admission and day 4, along with preadmission measurements of frailty, comorbidities, nutritional status and number of medications. Outcome measures included mortality and loss of autonomy on day 90 after admission.

### Results

Eighty-seven critically ill patients aged 80 years or older with preadmission functional independence and no missing SOFA score data on day 4 were studied (primary analyses). The mortality rate on day 90 was 30%. In a univariate Cox model, the SOFA score on day 4 was significantly associated with mortality rate: hazard ratio = 1.18 per one-point increase, 95% confidence interval (CI), 1.08 to 1.28 (p<0.001). A SOFA score of 6 or more on day 4 could correctly classify 75% of patients who died on day 90, with a sensitivity of 54% and a specificity of 84%. After adjustment, the SOFA score on day 4, neurological failure on admission and the number of preadmission medications were significantly associated with mortality on day 90, with an area under the receiver operating characteristic curve of 0.81 (95% CI, 0.71 to 0.91). These findings were confirmed in a sensitivity analysis with 109 patients. Preadmission frailty was the only variable independently associated with loss of autonomy in the 49 surviving patients.

**Data Availability Statement:** All relevant data are within the manuscript and its S1 Dataset files.

**Funding:** The authors received no specific funding for this work.

**Competing interests:** The authors have declared that no competing interests exist.

**Abbreviations:** ADL, Activities of Daily Living; AUC-ROC, area under the receiver operating characteristics; CI, confidence interval; CIRS-g, Cumulative Illness Rating Scale for Geriatrics; CSF, Clinical Frailty Score; GCS, Glasgow Coma Scale; HR, Hazard ratios; IADL, Instrumental Activities of Daily Living; MNA-SF, Mini Nutritional Assessment short form; OR, odds ratios; SOFA, sequential organ failure assessment; TLT, time-limited trial.

## Conclusion

Measuring SOFA score on day 4 and preadmission frailty could help predict mortality and loss of autonomy on day 90 in octogenarians after their acute admission to the ICU.

## Introduction

A growing number of patients aged 80 years or older are admitted to intensive care units (ICUs) [1, 2]. However, the benefits of an ICU stay are unclear for this population [1, 3, 4]. A recent program to promote systematic ICU admission among octogenarians found no major impact on mortality, functional status or quality of life at 6 months [5]. The risk of long-term death, i.e., 3 years after ICU discharge, was comparable between very old patients and the age- and sex-matched general population [6]. There were also controversial data about the impact of an ICU stay on functional status among survivors [7–9].

There is therefore a need to identify which octogenarians could benefit most from critical care. The decision to admit very old patients to the ICU is subject to high variability rates within centers [3]. Age cannot be the only decision-making factor; additional factors should be considered, such as the cause of admission to the ICU, comorbidities and/or preadmission frailty [10]. In addition, physicians may question the clinical relevance of continuing treatments for aged patients after their admission to the ICU. Measuring responses to treatments in the ICU has emerged as a strategy that helps clinicians to determine the trajectory of an individual's critical illness where medical uncertainty exists about the outcome. After a period of time, usually from 2 to 4 days, assessment of the response to therapies, as achieved with the sequential organ failure assessment (SOFA) [11], can objectively address whether or not there is an improvement in the patient's condition [12]. Changes in SOFA score were significantly associated with mortality in randomized controlled trials [13]. Whether dynamic SOFA score assessments could be indicative of outcome in octogenarians after their admission to the ICU is currently unknown. We aimed to see if SOFA scores on admission and on day 4 after admission, in conjunction with preadmission covariates, could be used as early indicators of mortality and/or dependency on day 90 after unplanned admission of octogenarians to the ICU.

## Methods

This prospective, observational cohort study was conducted between 15 January 2017 and 15 September 2018 with the participation of four adult ICUs of a French University Hospital: a surgical ICU (19 beds), a neurosurgical ICU (13 beds), a cardio-surgery ICU (20 beds) and a medical ICU (28 beds). The French ethical committee for the Clinical Research in Anesthesiology and Intensive Care (CERAR, IRB 00010254–2016–109) approved the study design on 1st October 2016 and, given its observational nature, waived the requirement for written informed consent from patients or relatives. Each patient or his/her relatives received oral and written information about the research, and they could refuse to participate according to French law [14].

### Participants

Patients were included if they were 80 years or older and had acute admission to the ICU for a foreseeable stay of more than 24 h in relation with severe infection and/or requirements of active therapies for organ failure on admission. Patients were not included if they stayed in the

ICU for less than 24 h, had an end-of-life decision before day 2 after admission, were admitted after elective surgery or during nights or weekends, had no French-language skills for telephone interviews, or if they or a relative opposed their participation in the study. If a patient was readmitted to the ICU during the study period, his/her first stay was considered.

## Data collection

With the registration to the National Institute of Health Data (INDS) (Ref#1894517p) on 19 August 2016, data acquired during the study period from unselected, consecutive patients were subsequently extracted from an ICU information management system (Centricity™ High Acuity Critical Care, GE Healthcare, Vélizy, France). Data included the patient's characteristics prior to ICU admission and ICU stay details recorded on admission day (day 1), on day 4 of the ICU stay and at discharge. Preadmission frailty was defined as a Clinical Frailty Score (CFS) of 5 or more [15]. The preadmission number of medications taken by the patient, including anticoagulants and antiplatelet agents, was collected. Comorbidities were assessed using the Cumulative Illness Rating Scale for Geriatrics (CIRS-g) [16]. The Mini Nutritional Assessment short form (MNA-SF) was used to assess nutritional status [17]. In the ICU, the severity of the medical condition was measured by determining the SOFA score on day 1 and day 4. The change in SOFA was then calculated ($\Delta$SOFA = SOFA on day 1 –SOFA on day 4): if the patient's condition improves, the change in SOFA score is positive, and vice-versa. Neurological failure on admission was defined as a Glasgow Coma Scale (GCS) score of 13 or less and/or spinal cord injury (quadriplegia, paraplegia) in the absence of sedation.

Patient status was assessed on day 90 after ICU admission during a telephone interview with the patient or a relative conducted by a trained physician (EL). During the interview, self-sufficiency and functional status on day 90 and prior to the ICU admission were assessed using the Katz Index of Independence in Activities of Daily Living (ADL) [18] and the Lawton Instrumental Activities of Daily Living (IADL) [19]. An ADL score of 6 indicated full independence, and a score of 2 or less high dependence on daily activities. Full functional activity was indicated with an IADL score of 8, and low functional activity with an IADL score of 0. Loss of autonomy (change in ADL) was calculated using ADL measurements on day 90 and on admission (= ADL on day 90 –ADL on day 1) and defined as a change in ADL score of less than 0.

## Statistical analysis

Data are expressed as median and interquartile range (25 to 75th quartiles), or number and percentage. To assess the SOFA score and change in SOFA score as possible variables associated with mortality on day 90, we fitted Cox regression models on the data from the subset of patients who were still in the ICU on day 4. We used the robust variance estimation method for the coefficients and relied on the Efron approximation of the exact marginal likelihood to handle tied survival times. The SOFA scores on day 1 and day 4, and change in SOFA score ($\Delta$SOFA) were tested after adjusting for mortality risk factors. These factors included patient characteristics prior to admission, organ failures and treatments on day 1 and day 4, and ICU characteristics. Factors were retained using a univariate regression model if they had a P value <0.20 and were tested for their independence. Hazard ratios (HR) and odds ratios (OR) with their 95% confidence intervals (95% CI) were computed from the estimated parameters of the final Cox and logistic regression models, respectively. SOFA scores and independent factors were tested for their diagnostic performance in correctly classifying survivors and non-survivors on day 90 using the area under the receiver operating characteristics (ROC) curve (AUC-ROC) (mean, 95% CI). In those patients still alive on day 90, the SOFA score at

admission and day 4 was tested as a potential risk factor for loss of autonomy using a logistic regression model after adjustment.

Analyses for survival and functional status were performed without imputing missing SOFA scores on day 4 (primary analysis). In the event of loss to follow-up before day 90, the last contact with the patient was considered. A sensitivity analysis was also conducted using imputed missing SOFA data on day 4 as follows: a SOFA score of 24 for patients who died between day 1 and day 3, the maximal SOFA score found in the whole cohort for patients who had an end-of-life decision between day 2 and day 4, and a SOFA score of 0 for patients who were discharged alive from the ICU before day 4. Missing data for covariates identified as adjustment factors were imputed using linear, Poisson or logistic regression predictions if the covariate was continuous, categorical or binary, respectively. No formal sample size calculation was performed for this observational study. However, on the basis of empirical investigations, a rule for sample size was to have at least 10 outcome events per parameter estimated [20]. Considering a mortality rate between 30% and 40% on day 90 in this population [3, 21], we estimated that a cohort of 100 patients would be enough to reliably determine 3–4 predictors using logistic regression analysis. Statistical significance was declared when $p \leq 0.05$ (Stata 15.1, Stata Corporation, College Station, TX, USA).

## Results

A total of 259 octogenarians had unplanned admission to an ICU during the study period. There were 150 non-included patients for reasons detailed in Fig 1. Other reasons for non-inclusion corresponded to patients admitted after elective surgery, admissions during nights, and unclear reasons for ICU admission. The analysis included 87 patients who stayed for 4 days or more in the ICU with no missing SOFA score data (primary analysis). The sensitivity analysis was conducted with 109 patients using imputed data for the SOFA score on day 4 accordingly: three patients with SOFA scores of 24, four patients with SOFA scores of 16, and 15 patients with SOFA scores of 0. Table 1 details the characteristics of the whole cohort. Most patients had functional independence prior to their admission. The main reasons for admission to the ICU were septic shock (n = 23), neurologic disorder (n = 23), acute respiratory failure (n = 17), hemorrhagic shock (n = 14), cardiac failure (n = 10), and multiple trauma (n = 10). Patients received vasoactive drugs (n = 62), respiratory support (n = 56), sedation (n = 16) and/or renal replacement therapy (n = 7) on admission.

### Survival at 3 months

Of the 87 patients included in the primary analysis, 55 patients were alive on day 90, 26 died during the study period (13 during the ICU stay) and six patients were lost to follow-up on day 90. The mortality rate on day 90 was 30%. The median time between admission and death was 15 days (7 to 47). On day 4, the median SOFA score was 3 (1 to 6) and the change in SOFA score was 2 (0 to 4).

In the univariate Cox model, the SOFA score on day 4 and on admission were significantly associated with mortality rate: HR = 1.18 per one-point increase, 95% CI, 1.08 to 1.28 (p<0.001), and HR = 1.10 per one-point increase, 95% CI, 1.00 to 1.22 (p = 0.053), respectively. The change in SOFA score was close to significance: HR = 1.14 per one-point increase, 95% CI, 0.99 to 1.31 (p = 0.059). After adjustment for imputed covariates, the SOFA score on day 4, neurological failure on admission and the number of preadmission medications were significantly associated with mortality (Table 2). The AUC-ROC curve of the SOFA score on day 4 associated with mortality on day 90 was 0.72 (95% CI, 0.60 to 0.84). A SOFA score of 6 or more on day 4 could correctly classify 75% of patients who subsequently died on day 90,

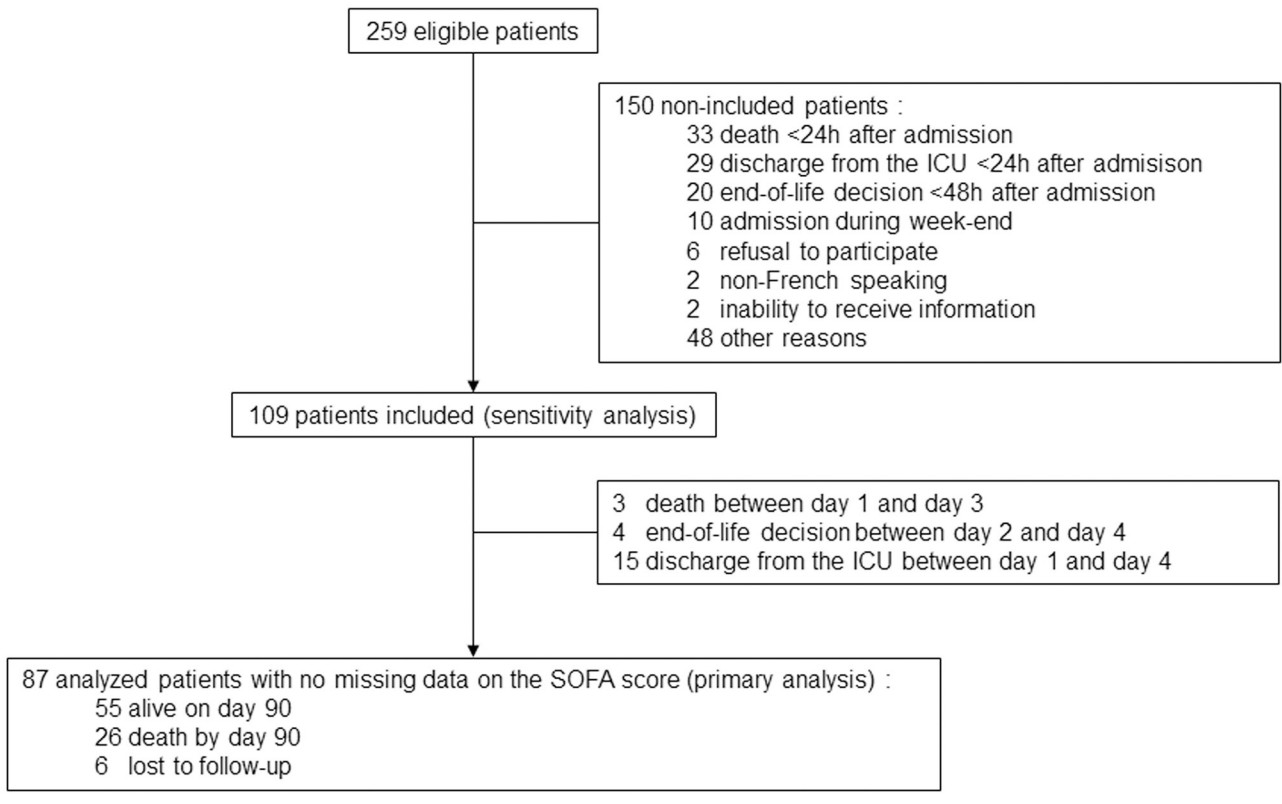

**Fig 1. Study flow diagram of patients.**

with a sensitivity of 54% and a specificity of 84%. When neurological failure on admission and the number of preadmission medications were added to the model, the AUC-ROC curve was 0.81 (95% CI, 0.71 to 0.91) (Fig 2).

Of the 109 patients in the sensitivity analysis, 7 patients were lost to follow-up on day 90. During the period of follow-up ranged from 74 to 160 days, 65 patients were alive at the last contact and 37 died (20 during the ICU stay). In univariate analysis, the imputed SOFA data, i.e., scores on day 1 and day 4 and change in SOFA score, were independently associated with mortality rate: HR = 1.11 per one-point increase, 95% CI, 1.01 to 1.22 (p = 0.029), HR = 1.27 per one-point increase, 95% CI, 1.20 to 1.35 (p<0.001) and HR = 1.28 per one-point increase, 95% CI, 1.21 to 1.36 (p<0.001), respectively. After adjustment, the SOFA score on day 4, change in SOFA score, neurological failure on admission and preadmission frailty were significantly associated with mortality (Table 3).

### Functional status on day 90

The follow-up on day 90 was possible for 49 survivors of the 87 patients. Their outcome was a return home (n = 37), continued hospitalization (n = 9) or transfer to an institution (n = 3). Median time to home return was 35 days (17 to 69). Fourteen patients were re-hospitalized after their first hospital discharge. The median ADL score was 6 (4 to 6) and change in ADL score was 0 (-2 to 0) on day 90. Thirty-one patients were independent in their daily activities (ADL 5 or 6) and six patients were dependent (ADL 2 or less). Overall, 30 survivors (61%) regained their previous functional status and six patients (12%) became dependent following

**Table 1. Baseline characteristics of patients (n = 109).** Data are expressed as median (25–75[th] percentiles) or number (%).

| Variables | Data |
|---|---|
| Age, years | 83 (81–86) |
| Male, n (%) | 55 (50) |
| Preadmission: | |
| CIRS-g score | 10 (7–14) |
| MNA-SF score[a] | 10 (6–12) |
| CFS ≥5, n (%)[b] | 48 (55) |
| Medications per patient, number[c] | 5 (3–8) |
| ADL score[d] | 6 (5–6) |
| IADL score[e] | 7 (4–8) |
| Most frequent comorbidities, n (%): | |
| Hypertension | 70 (64) |
| Cardiac | 57 (53) |
| Diabetes | 42 (39) |
| Respiratory | 28 (26) |
| Admission source, n (%): | |
| Medicine | 49 (45) |
| Surgery | 39 (36) |
| Trauma | 21 (19) |
| Type of ICU, n (%): | |
| Surgical ICU | 41 (38) |
| Medical ICU | 35 (32) |
| Cardiac ICU | 20 (18) |
| Neuro ICU | 13 (12) |
| SOFA score on admission | 6 (3–8) |
| SOFA score of 3 or 4 on admission[f], n (%) | |
| Cardiovascular | 59 (54) |
| Neurological | 23 (21) |
| Respiratory | 21 (19) |
| Renal | 13 (12) |
| Coagulation | 4 (4) |
| Hepatic | 2 (2) |
| Organ failure per patient on admission, number | 1 (0–2) |
| Duration in the ICU, days | 5 (3–9) |

CIRS-g, Cumulative Illness Rating Scale for Geriatrics; MNA-SF, Mini Nutritional Assessment short form; CSF, Clinical Frailty Score; ADL, Activities of Daily Living; IADL, Instrumental Activities of Daily Living; ICU, Intensive Care Unit; SOFA, sequential organ failure assessment.

[a]13 missing data;

[b]23 missing data;

[c]4 missing data;

[d]3 missing data;

[e]6 missing data.

[f]The number of individual organ failures exceeds the total number of included patients.

their ICU stay. Regarding patients' level of autonomy for carrying out instrumental activities, the median IADL score was 4 (4 to 8) and change in IADL score was -1 (-3 to 0) on day 90. Because ADL and IADL scores were strongly correlated, the IADL score was not considered in the analysis.

**Table 2. Multivariate Cox model of mortality on day 90 including SOFA scores adjusted for imputed and independent covariates (primary analysis with 87 patients).** The change in the SOFA score (ΔSOFA) was calculated as SOFA score on day 1 –SOFA score on day 4.

| Variables | Hazard ratio | 95% CI lower bound | 95% CI upper bound | P value |
|---|---|---|---|---|
| Antiplatelet agents | 0.79 | 0.26 | 2.43 | 0.685 |
| Anticoagulants | 1.19 | 0.37 | 3.79 | 0.768 |
| Preadmission medications | 1.14 | 1.03 | 1.26 | 0.013 |
| Neurological failure on admission | 3.53 | 1.24 | 10.04 | 0.014 |
| Respiratory support on day 4 | 0.74 | 0.25 | 2.19 | 0.586 |
| SOFA on day 4 | 1.17 | 1.01 | 1.35 | 0.033 |
| ΔSOFA | 1.13 | 0.95 | 1.36 | 0.167 |

SOFA, sequential organ failure assessment; CI, confidence interval.

SOFA scores on day 1 and day 4, and change in SOFA score were not significantly associated with loss of autonomy on day 90: HR = -0.05, 95% CI, -0.21 to 0.12 (p = 0.573), HR = -0.10, 95% CI, -0.31 to 0.11 (p = 0.372) and HR = -0.03, 95% CI, -0.26 to 0.19 (p = 0.788), respectively. In univariate analysis, loss of autonomy was significantly associated with preadmission frailty and CIRS-g score, ADL score and MNA-SF score at the time of ICU admission (all p<0.05). After adjustment, preadmission frailty was the only variable significantly associated with loss of autonomy: HR = 2.49, 95% CI, 0.06 to 4.92 (p = 0.045) (Table 4). No sensitivity analysis was performed due to the lack of effect of the SOFA score on functional outcome.

## Discussion

In a cohort of octogenarians, scoring SOFA on day 4 after unplanned admission to the ICU was the most representative factor associated with mortality on day 90. In addition, preadmission frailty was an indicator of loss of autonomy on day 90 in survivors. In clinical practice, these results indicate that very old age cannot be considered as the only decision-making factor to limit therapies in the ICU. Octogenarians may benefit from aggressive treatments if their initial condition can be objectively improved on day 4 after admission.

The decision to admit octogenarians to and/or undertake their care in the ICU is difficult. Physicians are reluctant to refer or admit very old patients to the ICU, even if they meet definite admission criteria [3, 22]. Older age is associated with higher rates of decisions to withhold ventilator support and renal replacement therapy [23]. To prevent the risk of self-fulfilling prophecies, studies have identified factors that can help physicians to decide which very old patients could benefit most from the ICU. Age, gender, preadmission frailty and SOFA score on day 1 were identified as independent factors of 30-day mortality after acute admission to the ICU [21]. Unplanned ICU admission [24], underlying fatal disease and severe functional limitation [25], acute and chronic renal failure [2], and preadmission frailty [21, 26, 27] were also associated with higher mortality rates.

Thereafter, discussion may occur about the clinical relevance of continuing therapies for very old patients admitted to the ICU. Among strategies for better selection was that of measuring patient response to therapy in the ICU. Response to therapy forms part of the concept of a time-limited trial (TLT) of treatment. TLT has been proposed to facilitate goal-concordant care in the specific environment of the ICU where medical uncertainty exists about the outcome. It represents an agreement between clinicians, a patient and his/her family to use therapies over a defined period of time and to see if the patient's condition improves or deteriorates according to defined outcomes [28, 29]. In critically ill patients with solid tumors, a TLT duration of 1 to 4 days after ICU admission using SOFA scores to assess the patient's condition was

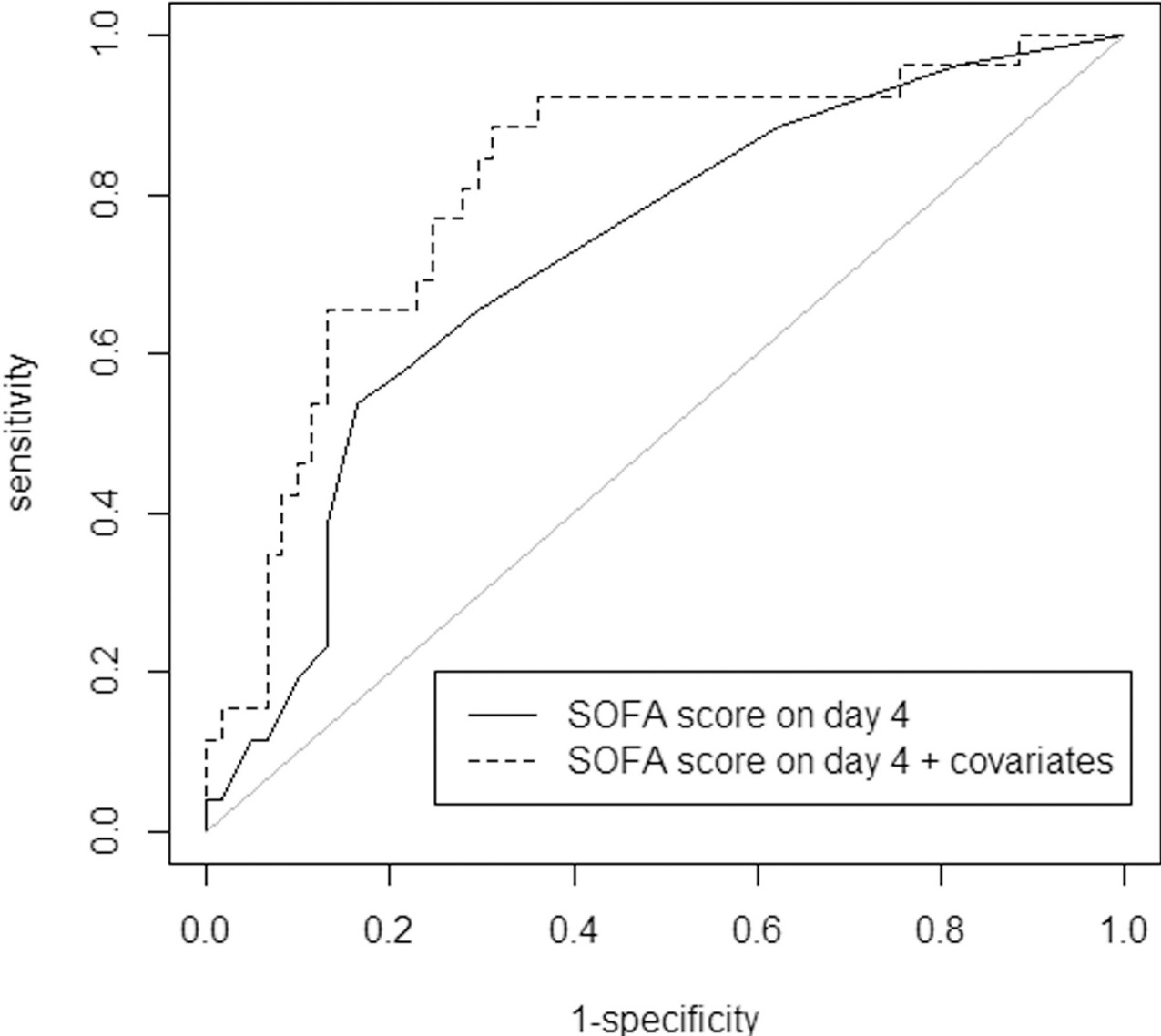

**Fig 2. Receiver operating characteristic (ROC) curves of predictive models for 3-month mortality in 87 critically ill octogenarians.** The *continuous line* represents the ROC curve of the sequential organ failure assessment (SOFA) score on day 4 after admission to the intensive care unit as a predictor of 3-month mortality. The *dashed line* represents the ROC curve of the multivariate Cox model including the SOFA score on day 4, neurological failure on admission and the number of preadmission medications as a predictor of 3-month mortality. The area under the ROC curve (AUC-ROC) of the SOFA score on day 4 was 0.72 (95% CI, 0.60 to 0.84), and the AUC-ROC curve of the SOFA score on day 4 with covariates added was 0.81 (95% CI, 0.71 to 0.91).

found to be sufficient to achieve the optimal survival benefit [30]. However, TLTs are conducted infrequently in the ICU [29]. The present study suggests that measuring SOFA scores on day 4 could be incorporated into the TLT concept for octogenarians in the ICU.

We found that the SOFA score on day 4 was associated with mortality on day 90. The median change in SOFA score was 2 (0 to 4), suggesting that a positive impact of therapies could be objectively seen within 4 days after admission. Although we did not model the optimal timing for measuring the SOFA score, the choice of day 4 after admission was based on literature [30, 31] and our clinical experience. Using machine learning, predictive power of

**Table 3. Multivariate Cox model of mortality on day 90 including imputed SOFA scores adjusted for imputed and independent covariates (sensitivity analysis with 109 patients).** The change in the SOFA score (ΔSOFA) was calculated as SOFA score on day 1 –SOFA score on day 4.

| Variables | Hazard ratio | 95% CI lower bound | 95% CI upper bound | P value |
|---|---|---|---|---|
| Antiplatelet agents | 1.04 | 0.41 | 2.66 | 0.927 |
| Preadmission frailty | 2.54 | 1.21 | 5.33 | 0.014 |
| Neurological failure on admission | 2.35 | 1.06 | 5.21 | 0.036 |
| SOFA on day 4 | 1.26 | 1.14 | 1.39 | <0.001 |
| ΔSOFA | 1.10 | 1.00 | 1.20 | 0.050 |

SOFA, sequential organ failure assessment; CI, confidence interval.

mortality was maximal on the second day of admission for the general population of the ICU [32]. Interestingly, we found that the SOFA score measured on admission was less informative for predicting outcomes than the SOFA score on day 4, possibly because the early response to therapies was not incorporated into this risk score derived from cohort studies on middle-aged patients [11].

The predictive performance of SOFA measurements on day 4 was further increased when neurological failure on admission and the number of preadmission medications were considered. Neurological failure on admission was defined as disorders of consciousness and/or spinal cord injury. Because spinal cord injury is associated with higher mortality rates in older patients than in younger patients [33], this condition should be considered in addition to the GCS score contained in the SOFA score. Concerning the number of preadmission medications, a causal relationship between polypharmacy and mortality in very old patients remains unclear [34, 35].

Preadmission frailty was a factor associated with mortality (sensitivity analysis) and loss of autonomy (primary analysis). Frailty as assessed with a CFS score of 5 or more was associated with higher short-term mortality in very old ICU patients [21, 26, 27]. However, there was no impact of this scoring on in-hospital mortality in ICU patients with pneumonia [36]. Its use as the sole component to determine access of old patients to health care is debated [37]. Our findings indicate that frailty was not a major indicator of mortality, but was closely associated with loss of autonomy on day 90. Frailty has five components: physical impairment, psychological

**Table 4. Logistic regression model of loss of autonomy for patients who were still alive on day 90 including SOFA scores adjusted for independent covariates (primary analysis with 49 patients).** The loss of autonomy (change in [Δ] ADL) was calculated as ADL score at 90 days–ADL score on day 1, and was defined as ΔADL <0. The change in the SOFA score (ΔSOFA) was calculated as SOFA score on day 1 –SOFA score on day 4.

| Variables | Odds ratio | 95% CI lower bound | 95% CI upper bound | P value |
|---|---|---|---|---|
| Age | 0.08 | -0.19 | 0.35 | 0.541 |
| Preadmission frailty | 2.49 | 0.06 | 4.92 | 0.045 |
| Preadmission CIRS-g | 0.14 | -0.11 | 0.38 | 0.274 |
| Preadmission medications | 0.30 | -0.13 | 0.73 | 0.168 |
| Preadmission anticoagulant | -0.85 | -3.68 | 1.99 | 0.559 |
| ADL score on admission | -0.70 | -3.02 | 1.63 | 0.556 |
| Neurological failure on admission | -0.39 | -2.33 | 1.55 | 0.694 |
| Respiratory support on admission | -0.90 | -3.28 | 1.48 | 0.458 |
| SOFA on day 4 | -0.21 | -0.59 | 0.18 | 0.302 |
| ΔSOFA | 0.21 | -0.25 | 0.67 | 0.369 |

SOFA, sequential organ failure assessment; CI, confidence interval; CIRS-g, Cumulative Illness Rating Scale for Geriatrics; ADL, Activities of Daily Living.

impairment, cognitive impairment, financial limitation and limited social support [38]. Due to its global content, CFS looks appropriate to estimate frailty in very old ICU patients. Of note was that other scales that can measure geriatric syndromes on admission, such as ADL, IADL, nutritional status and comorbidities, did not improve the model, as previously noted [27].

Our study has several limitations. First, the study was conducted in four sites of one university hospital. Whether the identified variables in the models can be transposed to other sites warrants further investigation. Second, we identified variables associated with outcome in this cohort. A validation cohort is however required prior to considering their use in a decision-making process. Third, included patients had no end-of-life decision taken within 24 h of their admission. However, we cannot exclude that decisions to withdraw and withhold life-sustaining therapies were taken after day 4. Of note was the in-ICU mortality of 50% of the whole mortality, suggesting that a decision to limit life-sustaining therapies in the ICU had not been made for a substantial number of non-survivors. Fourth, the number of included patients is limited despite the participation of four ICUs for a 20-month study period. This may reflect the restrictive access of octogenarians to the ICU. In addition, the limited number of patients could have affected the determination of clinically relevant variables in the models. These issues indicate that our results must be interpreted with caution.

In conclusion, measuring SOFA scores on day 4 after the admission to the ICU was significantly associated with mortality on day 90 in octogenarians. The question about the clinical relevance of continuing therapies in the ICU should incorporate measurements of early responses to therapy should these patients be admitted to the ICU.

## Supporting information

**S1 Dataset.**
(XLSX)

## Acknowledgments

The authors wish to thank Drs Marie-Christine Herault, Michel Durand, and Clotilde Schilte (surgical ICU, cardio-surgical ICU and neurosurgical ICU, respectively) for their help in recruiting participants for the study. The authors thank also Dr. Pierre Gillois (Department of Public Health, Grenoble Alpes University Hospital, France) for his contribution to the data analysis.

## Author Contributions

**Conceptualization:** Emmanuelle Loyrion, Thibaut Trouve-Buisson, Gaetan Gavazzi, Jean-Luc Bosson, Jean-François Payen.

**Data curation:** Emmanuelle Loyrion, Thibaut Trouve-Buisson, Carole Schwebel.

**Formal analysis:** Emmanuelle Loyrion, Lydiane Agier, Carole Schwebel, Jean-Luc Bosson, Jean-François Payen.

**Investigation:** Emmanuelle Loyrion, Thibaut Trouve-Buisson, Gaetan Gavazzi, Carole Schwebel.

**Methodology:** Lydiane Agier, Jean-Luc Bosson.

**Supervision:** Jean-François Payen.

**Validation:** Gaetan Gavazzi.

**Writing – original draft:** Emmanuelle Loyrion, Lydiane Agier, Jean-François Payen.

**Writing – review & editing:** Thibaut Trouve-Buisson, Gaetan Gavazzi, Carole Schwebel, Jean-Luc Bosson.

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
