## [Decision Letter · Decision Letter 0]

11 May 2021

PONE-D-21-10444

Which early indicators associated with outcomes in very old critically ill patients?

PLOS ONE

Dear Dr. Payen,

Thank you for submitting your manuscript to PLOS ONE. After careful consideration, we feel that it has merit but does not fully meet PLOS ONE’s publication criteria as it currently stands. Therefore, we invite you to submit a revised version of the manuscript that addresses the points raised during the review process.

 ACADEMIC EDITOR: 

The manuscript presents an intersting and relavant analysis. But the population is small and eterogeneous and there is not a sample size calculation. So the results should be interpreted with caution. 

Which type of study is this? cross-sectional study as defined in the abstract or a prospective observational cohort study? 

Some important informations about the reason for Admission and first therapy are also not so clear.

There are no conflicts between the reviews so that it's clear which advice the authors should follow.

Please authors should answer to all the questions moved by the reviewers, to make the manuscript suitable for publication.

We look forward to receiving your revised manuscript.

Kind regards,

Martina Crivellari

Academic Editor

PLOS ONE

Journal Requirements:

2. PLOS ONE does not copy edit accepted manuscripts (https://journals.plos.org/plosone/s/criteria-for-publication#loc-5). To that effect, please ensure that your submission is free of typos and grammatical errors, including in the title.

Reviewers' comments:

Reviewer's Responses to Questions

**Comments to the Author**

1. Is the manuscript technically sound, and do the data support the conclusions?

Reviewer #1: Partly

Reviewer #2: Partly

2. Has the statistical analysis been performed appropriately and rigorously? 

Reviewer #1: Yes

Reviewer #2: Yes

3. Have the authors made all data underlying the findings in their manuscript fully available?

Reviewer #1: Yes

Reviewer #2: Yes

4. Is the manuscript presented in an intelligible fashion and written in standard English?

Reviewer #1: Yes

Reviewer #2: Yes

5. Review Comments to the Author

Reviewer #1: The study "Which early indicators associated with outcomes in very old critically ill patients? presents an interesting Analysis on older patients admitted to ICU and Outcome prediction.

The study is intersting a relevant but I'd like to highlight a number of issues thet should be adressed be the authors:

The title should include "are".

1)I would suggest to use "octogenarians" intead of "very old patients".

2)How could a stay longer than 24 be predicted?

3) It is not clear how a "end of life" decision before 48 Hours excluded patients from the study and not when taken later during the Hospital stay.

4) The authors should specify the reason for Admission. When they stated Surgery, does it mean post surgical? Was it post emergency or planned surgery?

5) The Population is relatively small and Comes from four very different ICUS.

6) Th ereason for Admission to ICU may also have played a role an certaily would help to direct the Interpretation of the results. What was the main failure or combination of filures? What was the Treatment initiated (e.g. mechanical Ventilation/Dialysis)

7) I appreciated the idea of TLT. Did the Authors looked at SOFA Change and absolute number at 24 or 48 Hours? Would they expect them to be different?

7) Was there any power Analysis done?

8) How would this results practically Impact on clinical decision making?

Reviewer #2: Thank you for asking me to review this exciting manuscript.

Summary

Payen et al. evaluated the association of early indicators, i.e., the SOFA score at admission (day 1) and day four, along with other factors, with long-term outcomes, i.e., 90-day mortality and functional status among ≥80-year-old patients. They found that the combination of day 4 SOFA score and other factors associated with mortality increased the prediction of death as evidenced by the AUROC of 0.81 compared to the only SOFA at day four is 0.72. In addition, they found that SOFA at day four, unlike at admission, was significantly associated with mortality. However, this asserts the message that predictive tools should not be used in isolation to determine outcomes for individual patients. Provision of a time-limited trial is, therefore, could potentially be justified; however, these results ought to be interpreted with caution because of the petite sample size.

While there are ample studies on elderly patients and factors associated with their outcomes, both short- and long-term, this study focuses on using the SOFA score akin to a time-limited trial to assess association with long-term outcomes. While it’s not novel but highlights an important message, due to some of the methodological challenges, the validity of the results is in question, especially related to the functional assessment.

Major comments

1. Methods

a) The authors should highlight the sampling method used for this study.

b) How did the authors arrive at the sample size they needed? There was no sample size calculation description made. The authors should perhaps give the reason for this. Sample size is essential for a study whose aim is to assess the predictive ability of tools like the SOFA score and other factors.

c) The function assessment was both with regards to independence (self-sufficiency) and functional activity. However, the Katz Index of Independence in Activities of Daily Living (ADL) assesses independence, while the Lawton Instrumental Activities of Daily Living (IADL) assess functional activity. However, in lines 121-124, the authors indicate that the tools were assessing the opposite i.e., “An ADL score of 6 indicated full function, and a score of 2 or less severe functional impairment. Full independence was indicated with an IADL score of 8 and dependence with an IADL score of 0.” Perhaps the authors can provide clarification to this.

Minor

1. General comments

a) Some minor grammatical errors need to be fixed especially in the results section as well as clarity of writing for the reader to grasp the message e.g. Figure 1, “End-of-decision < 48h after admission”

2. Title

a) The title doesn’t highlight the key aim of the research, which is to use SOFA score in conjunction with other variables in predicting long-term outcomes of very old critically ill patients.

b) Perhaps the authors should consider highlighting the SOFA score in the title and including the study design in the title.

3. Abstract

a) However, the abstract refers to it as a cross-sectional study (line 33), while the methods section, line 82, refers to it as a prospective observational cohort study. The authors should clarify this discrepancy; however, this is a prospective cohort study by design.

4. Methods

a) It is clear how long the follow-up period was, i.e., 90 days after ICU admission; however, the authors should include how long the recruited patients. When did the recruitment end?

b) It is unclear why night or weekend admissions were excluded.

c) In Figure 1, Exclusion based on “Unknown reason” is unclear. Authors perhaps should explain these criteria with some examples if possible in the Results section.

5. Limitations

a) Due to the petite sample size, the authors clearly state that their results ought to be interpreted with caution.

6. PLOS authors have the option to publish the peer review history of their article (what does this mean?). If published, this will include your full peer review and any attached files.

Reviewer #1: **Yes: **Massimiliano Meineri

Reviewer #2: No

---

## [Author Response · Author response to Decision Letter 0]

26 May 2021

POINT-BY-POINT RESPONSE TO REVIEWERS 

We thank the reviewers for their constructive criticisms and suggestions that have helped us to clarify the paper and to enhance the quality of the message.

Academic Editor

The manuscript presents an interesting and relevant analysis. But the population is small and heterogeneous and there is not a sample size calculation. So the results should be interpreted with caution. 

Which type of study is this? cross-sectional study as defined in the abstract or a prospective observational cohort study? 

Some important information about the reason for Admission and first therapy are also not so clear.

ANSWER: We thank you for these comments. These issues have been addressed accordingly. Please see our responses below to Rev#1 and 2.

 

Reviewer #1

The study "Which early indicators associated with outcomes in very old critically ill patients?” presents an interesting Analysis on older patients admitted to ICU and Outcome prediction.

The study is interesting a relevant but I'd like to highlight a number of issues they should be addressed be the authors.

The title should include "are".

ANSWER: As suggested by Rev#2, the title has been changed.

1) I would suggest to use "octogenarians" instead of "very old patients".

ANSWER: Done.

2) How could a stay longer than 24 be predicted?

ANSWER: More information about this point has been provided now (lines 95-96). 

3) It is not clear how a "end of life" decision before 48 Hours excluded patients from the study and not when taken later during the Hospital stay.

ANSWER: We have considered that early EOL decision was due to the extreme severity of illness on admission. This situation was similar to that of moribund patients staying in the ICU for less than 24h. However, EOL decisions could have been taken later if the medical team deemed that the patient’s condition worsened despite active therapies. These patients were then included in the study either in the primary analysis where EOL decision was taken after day 4 or in the sensitivity analysis with EOL decision between day 2 and day 4 (see Fig.1). 

4) The authors should specify the reason for Admission. When they stated Surgery, does it mean post surgical? Was it post emergency or planned surgery?

ANSWER: As mentioned in the MM section (line 98), patients were not included if they were admitted after elective surgery. The “unknown reason” term has been changed to “other reasons” with more information (lines 167-169). 

5) The Population is relatively small and comes from four very different ICUS.

ANSWER: This comment is right. This issue has been more discussed as a limit of the study (lines 338-342). 

6) The reason for Admission to ICU may also have played a role and certainly would help to direct the Interpretation of the results. What was the main failure or combination of failures? What was the Treatment initiated (e.g. mechanical Ventilation/Dialysis).

ANSWER: More information on the reasons for admission is provided now (lines 174-177). The most frequent organ failures and the most required treatments have been mentioned (please see Table 1 and lines 177-178). 

7) I appreciated the idea of TLT. Did the Authors looked at SOFA Change and absolute number at 24 or 48 Hours? Would they expect them to be different?

ANSWER: Unfortunately, we could not assess SOFA score on day 2 and day 3. However, serial measurements of SOFA score have been rarely performed in the literature. The use of changes in SOFA scores (delta SOFA) to predict mortality has been recently discussed (lines 75-76).

8) Was there any power Analysis done?

ANSWER: As requested also by Rev.2, more information is provided regarding the estimation of the sample size (lines 157-162).

9) How would this results practically Impact on clinical decision making?

ANSWER: We have added a summary statement from our results in the Discussion (lines 273-276).

 

Reviewer #2

Thank you for asking me to review this exciting manuscript.

Major comments

1. Methods

a) The authors should highlight the sampling method used for this study.

ANSWER: Data from unselected, consecutive patients were extracted during the study period. This point is mentioned now (lines 105-106).

b) How did the authors arrive at the sample size they needed? There was no sample size calculation description made. The authors should perhaps give the reason for this. Sample size is essential for a study whose aim is to assess the predictive ability of tools like the SOFA score and other factors.

ANSWER: This comment is right. As requested also by Rev.1, more information is provided regarding the estimation of the sample size (lines 157-162). 

c) The function assessment was both with regards to independence (self-sufficiency) and functional activity. However, the Katz Index of Independence in Activities of Daily Living (ADL) assesses independence, while the Lawton Instrumental Activities of Daily Living (IADL) assess functional activity. However, in lines 121-124, the authors indicate that the tools were assessing the opposite i.e., “An ADL score of 6 indicated full function, and a score of 2 or less severe functional impairment. Full independence was indicated with an IADL score of 8 and dependence with an IADL score of 0.” Perhaps the authors can provide clarification to this.

ANSWER: This suggestion is correct. Wording regarding these scores has been changed accordingly (lines 125-126).

Minor

1. General comments

a) Some minor grammatical errors need to be fixed especially in the results section as well as clarity of writing for the reader to grasp the message e.g. Figure 1, “End-of-decision < 48h after admission”

ANSWER: Please note that the manuscript had been reviewed and corrected by an English professional medical writer, who was not involved in the study. We have double-checked the material to prevent grammatical errors. Fig.1 has been corrected accordingly.

2. Title

a) The title doesn’t highlight the key aim of the research, which is to use SOFA score in conjunction with other variables in predicting long-term outcomes of very old critically ill patients.

ANSWER: The title has been changed as suggested.

b) Perhaps the authors should consider highlighting the SOFA score in the title and including the study design in the title.

ANSWER: Done. Due to the length of the new title, it was not possible to insert the study design.

3. Abstract

a) However, the abstract refers to it as a cross-sectional study (line 33), while the methods section, line 82, refers to it as a prospective observational cohort study. The authors should clarify this discrepancy; however, this is a prospective cohort study by design.

ANSWER: Abstract corrected.

4. Methods

a) It is clear how long the follow-up period was, i.e., 90 days after ICU admission; however, the authors should include how long the recruited patients. When did the recruitment end?

ANSWER: As mentioned in the MM section, the recruitment ended on 15 September 2018 (line 84). The status of the latest enrolled patient was assessed on day 90 after his/her admission, i.e. 15 January 2019. 

b) It is unclear why night or weekend admissions were excluded.

ANSWER: Data had to be collected on real-time, in particular those regarding preadmission frailty and nutritional status. To do so, only one person (EL) was in charge of this task to ensure their reliability and consistency. This explains why inclusions during nights or weekends were not possible. 

c) In Figure 1, Exclusion based on “Unknown reason” is unclear. Authors perhaps should explain these criteria with some examples if possible in the Results section.

ANSWER: The “unknown reason” term has been changed to “other reasons” with more information (lines 167-169).

5. Limitations

a) Due to the petite sample size, the authors clearly state that their results ought to be interpreted with caution.

ANSWER: This comment is right. This issue has been more discussed as a limit of the study (lines 338-342).

---

## [Editor Report · Decision Letter 1]

28 May 2021

Dynamic SOFA score assessments to predict outcomes after acute admission of octogenarians to the intensive care unit

PONE-D-21-10444R1

Dear Dr. Payen,

We’re pleased to inform you that your manuscript has been judged scientifically suitable for publication and will be formally accepted for publication once it meets all outstanding technical requirements.

Kind regards,

Martina Crivellari

Academic Editor

PLOS ONE
---

## [Editor Report · Acceptance letter]

22 Jul 2021

PONE-D-21-10444R1 

Dynamic SOFA score assessments to predict outcomes after acute admission of octogenarians to the intensive care unit 

Dear Dr. Payen:

I'm pleased to inform you that your manuscript has been deemed suitable for publication in PLOS ONE. Congratulations! Your manuscript is now with our production department. 

Kind regards, 

on behalf of

Dr. Martina Crivellari 

Academic Editor

PLOS ONE